# Ethanol to Acetaldehyde Conversion under Thermal and Microwave Heating of ZnO-CuO-SiO_2_ Modified with WC Nanoparticles

**DOI:** 10.3390/molecules26071955

**Published:** 2021-03-31

**Authors:** Alexander L. Kustov, Andrey L. Tarasov, Olga P. Tkachenko, Igor V. Mishin, Gennady I. Kapustin, Leonid M. Kustov

**Affiliations:** 1N.D. Zelinsky Institute of Organic Chemistry, Leninsky Prospect 47, 119991 Moscow, Russia; kyst@list.ru (A.L.K.); atarasov@ioc.ac.ru (A.L.T.); ot@ioc.ac.ru (O.P.T.); igo@ioc.ac.ru (I.V.M.); gik@ioc.ac.ru (G.I.K.); 2Chemistry Department, Lomonosov Moscow State University, Leninskie Gory 1, bldg. 3, 119992 Moscow, Russia; 3Institute of Ecology and Engineering, National University of Science and Technology “MISiS”, Leninsky Prospect 4, 119991 Moscow, Russia

**Keywords:** catalysis, acetaldehyde, copper oxide, ethanol, dehydrogenation, tungsten carbide

## Abstract

The nonoxidative conversion of ethanol to acetaldehyde under thermal and microwave heating was studied on mixed oxide ZnO-CuO-SiO_2_ catalysts modified with additives of tungsten carbide nanoparticles. The results revealed that the WC-modified catalyst exhibited superior activity and selectivity under microwave heating conditions. It is assumed that when microwave heating is used, hot zones can appear at the contact points of WC nanoparticles and active centers of the mixed oxide ZnO-CuO-SiO_2_ catalyst, which intensively absorb microwave energy, allowing the more efficient formation of acetaldehyde at moderate temperatures. Thermodynamic calculations of equilibrium concentrations of reagents and products allowed us to identify the optimal conditions for effective acetaldehyde production. The initial catalyst and the catalyst prepared by the coprecipitation of the oxides with the addition of WC were characterized by physicochemical methods (TPR-H_2_, XRD, DRIFTS of adsorbed CO). The active centers of the oxide catalyst can be Cu^+^ cations.

## 1. Introduction

Acetaldehyde (AA) is one of the most important intermediate aliphatic chemicals serving as the raw material for the production of acetic acid, acetic anhydride, ethyl acetate, pyridine and many other products [1]. The process of AA production by catalytic nonoxidative dehydrogenation of ethanol widely used in the 1960s–1970s of the last century has a number of advantages over other methods, such as acetylene hydration, ethylene oxidation and oxidative dehydrogenation of ethanol. These include the absence of toxic wastes, sufficiently mild reaction conditions and the production of hydrogen along with acetaldehyde, which can be used in other processes [2]. The starting material of this process is ethanol, the production of which from biomass (bioethanol) allows avoiding the use of petroleum-derived raw materials.

The most common and effective catalyst for the dehydrogenation of alcohols is copper, mainly due to its ability to preserve unbroken C-C bonds. The results of studies of copper-containing catalysts indicate that copper (Cu^0^) in the metallic state formed during the reduction of CuO is an active catalyst in the dehydrogenation reaction [3]. Ethanol dehydrogenation reaction with copper catalysts based on various oxides of SiO_2_, ZrO_2_, Al_2_O_3_, MgO and ZnO have been studied by many authors [4]. Among AA, a number of byproducts were found. Thus, ZrO_2_ and ZnO oxides contributed to the formation of ethyl acetate and acetone, respectively; Cu/ZrO_2_ and Cu/ZnO showed a maximum selectivity in the formation of ethyl acetate (27.6% and 28.1%). The highest selectivity for acetaldehyde was achieved in the presence of Cu/SiO_2_ and Cu/MgO catalysts (77.9% and 74.2%, respectively).

Some authors [5] concluded that copper-containing catalysts based on SiO_2_ are the most effective and selective. The advantages of such a carrier include a large specific surface area and a uniform distribution of copper particles on the surface of the support. Already at 275 °C, the conversion of ethanol was 77% with the selectivity for acetaldehyde of about 100%. However, these catalysts were unstable and deactivated after 2–3 h due to the sintering of copper particles.

At present, mixed oxide, zinc–copper–aluminum systems are the most promising catalysts for the production of AA by the ethanol dehydrogenation reaction. An industrial ZnO-CuO-Al_2_O_3_ (CZA) catalyst for the synthesis of methanol, as well as its mechanical mixtures with ZrO_2_, CeO_2_, Al_2_O_3_ and SiO_2_, were used as catalysts [6]. Gas mixtures significantly diluted with nitrogen (12 vol% of ethanol) were used in the reaction. It was shown that the selectivity of the CZA catalyst without additives reaches 95%; however, the productivity was no more than 2.2 μmol/(g_cat_ h). For the samples modified with additives of various oxides, the productivity increased to 5.7 μmol/(g_cat_ h), with considerable amounts of ethyl acetate appearing in the reaction products (up to 40 wt%).

From the point of view of reducing energy consumption, especially for reactions involving polar molecules such as ethyl alcohol, the use of microwave heating is topical because not only the catalyst but also the raw materials are heated. In our recent review [7], examples are given of the use of microwave radiation, which causes an increase in the activity and selectivity of various heterogeneous catalytic gas-phase reactions. The conversion of ethanol to divinyl is reported as well [8]. At the same time, there are no methods for performing a microwave-activated reaction of the nonoxidative dehydrogenation of ethanol into AA in the literature. This is because most copper-containing catalysts that are promising in the ethanol dehydrogenation reaction, including mixed oxides, ZnO-CuO-Al_2_O_3_, are practically not heated under the action of microwave energy even in the reduced state because of the low dielectric loss tangent of these materials.

There is a method allowing one to heat up such materials to high temperatures, which consists of the use of mechanical catalyst mixtures with materials capable of intensively absorbing microwave energy, such as carbon materials, carbides of various metals or titanium oxide. In our previous work [9], a Mn-Na_2_WO_4_/SiO_2_ catalyst of methane oxidative condensation was mixed (ground) in a ball mill with tungsten carbide, which allowed heating in the microwave field to 750–800 °C and increasing the yield and selectivity for ethylene. This approach is worthwhile to apply in studying the dehydrogenation of ethanol under microwave heating of ZnO-CuO-SiO_2_ catalysts, which are quite promising in this process. The use of tungsten carbide as a microwave-absorbing agent is justified by a rather high tangent of dielectric loss of WC that provides improved conditions for energy exchange with the ZnO-CuO-SiO_2_ catalyst that has a negligible value of the tangent of dielectric loss. Moreover, the application of nanoparticles of WC further improves the efficiency of heating due to the closer contacts of nanosize MW-absorbing particles with the active catalyst.

This manuscript is a continuation of an article previously published by one of the authors (A. L. Tarasov) [10]. The present manuscript deals with the study of the catalytic system, ZnO-CuO-SiO_2_ modified with WC, and the previous article focused on the study of the ZnO-CuO-SiO_2_ oxide catalysts modified with additives of Nb and Ta carbides. Both catalytic systems are used in the ethanol to acetaldehyde conversion activated by microwave radiation. This manuscript additionally includes extensive characterization results, including TPR-H_2_, XRD and DRIFTS of adsorbed CO, whereas the previous article only presented catalytic activity data.

Thus, the aim of this work was a comparative study of the catalytic properties of mixed oxide ZnO-CuO-SiO_2_ catalysts modified at the stage of their preparation with tungsten carbide (WC) in the reaction of ethanol to AA conversion under thermal and microwave heating conditions. While analyzing the results and choosing the conditions for carrying out the reaction, thermodynamic calculations of the equilibrium concentrations of products and reagents were used.

## 2. Results and Discussion

The composition and specific surface area of the catalysts are shown in Table 1. It should be noted that the mass ratio of zinc, copper and silicon oxides in the samples of these catalysts is the same.

### 2.1. Features of Microwave Heating

We studied various additives except for carbides of W, Nb and Ta, such as carbon materials, V-Mo oxide systems, SiC, etc., with the goal to identify the material that (1) absorbs microwave radiation very well, (2) does not provide any catalytic activity in the chosen reaction and (3) is quite stable under the experimental conditions of the catalytic reaction. WC was found to fit all three criteria; moreover, it was a better microwave absorber than TaC and NbC, and therefore, it was chosen for the detailed study. It was found that adding about 10 wt % of the WC nanopowder to the mixed oxide ZnO-CuO-SiO_2_ catalyst allows the catalyst to be heated by microwave energy to temperatures (200–400 °C) required for the nonoxidative dehydrogenation of ethanol. It should be noted that the WC nanopowder alone used under the action of microwave radiation with a power of 800 W is heated in a nitrogen stream to substantially higher temperatures (about 850–900 °C).

### 2.2. Results of Thermodynamic Calculations

The equilibrium composition of the gas mixture in the reaction of nonoxidative dehydrogenation of ethanol to form acetaldehyde (C_2_H_4_O) and ethyl acetate (C_4_H_8_O_2_) as a function of temperature, calculated using the computer program HSC-4, was reported in our previous paper [10]. It was shown that in addition to the main reaction of the formation of acetaldehyde (AA), a secondary reaction of formation of ethyl acetate (EA) takes place by dehydrocondensation of the starting ethyl alcohol and the resulting acetaldehyde: C_2_H_6_O + C_2_H_4_O → C_4_H_8_O_2_ + H_2_ (ΔH = −43.45 kJ mol^−1^). This reaction is exothermic and thermodynamically favorable; therefore. EA is a more advantageous product at low temperatures, whereas the selective formation of AA is favored by higher temperatures (above 400 °C). The formation of EA is also possible according to the Cannizzaro reaction (disproportionation of aldehyde): C_2_H_4_O → C_4_H_8_O_2_ + C_2_H_6_O. Based on these data, a sufficiently high mass flow rate of ethanol (3 h^−1^) was used to reduce the contribution of the secondary reaction and thereby increase the selectivity for AA.

### 2.3. Catalytic Results under Thermal and Microwave Heating

Table 2 shows the experimental values of the conversion and selectivity for AA at different reaction temperatures under thermal heating conditions of the initial unmodified catalyst sample and the calculated values of these parameters based on the reaction thermodynamics, according to the thermodynamic analysis.

The values of the alcohol conversion and AA selectivity at the high temperature (400 °C) obtained under thermal heating conditions are in good agreement with the results of thermodynamic calculations (Table 2). In the moderate temperature range (200–300 °C), the experimental values of the alcohol conversion are much lower than the calculated values, and the selectivities for AA are slightly higher than the calculated ones. A similar dependence was observed on other samples synthesized by us and modified with additives of Nb and Ta carbides. The results of the research will be published separately. In our opinion, this is due to the short contact time at the used high mass feed rate of ethanol (3 h^−1^). It should be noted that, in our case, in addition to AA, ethyl acetate and hydrogen are formed in the reaction, in agreement with the data obtained on a mixed oxide ZnO-CuO-Al_2_O_3_ catalyst [6].

The dependence of the conversion of ethanol and the selectivity of AA formation on the reaction temperature under thermal and microwave heating for WC-modified catalysts are shown in Figure 1.

Figure 1 shows that for the carbide-modified sample, the use of microwave heating leads to a significant increase in the ethanol conversion and the selectivity of AA formation in the region of low (200–300 °C) temperatures. At elevated (400 °C) temperatures, the conversion of ethanol and the selectivity of AA formation are practically independent of the heating method. Since microwave heating of the catalyst leads to a simultaneous growth of both activity and selectivity, it is natural that this is most strongly reflected in the performance of catalysts in AA formation. Thus, at 200–300 °C, under microwave heating conditions, the AA productivity (μmol_AA_/(g_cat_ h)) of the sample modified with tungsten carbide is higher than in the case of thermal heating by a factor of 1.5–3. It can be assumed that the use of microwave heating causes the appearance of “hot” zones in the contact points between the active ZnO-CuO-SiO_2_ catalyst and the catalytically inert modifier (WC nanopowder) that intensively absorbs microwave energy. As a result, this provides the more efficient formation of AA at low and moderate temperatures. It should be noted that the productivity obtained at low temperatures (200 °C) by other authors [6], up to 5.7 μmol_AA_/(g_cat_ h) for ethanol converted on similar catalysts with a ZnO-CuO-Al_2_O_3_ composition, with the addition of promoting additives (SiO_2_, CeO_2_ and ZrO_2_) is much lower than for the catalyst (ZnO-22.8, CuO-58.8, SiO_2_-9.2, WC-9.1) synthesized by us (Figure 1). This is especially noticeable for the reaction under microwave heating.

Particular attention should be paid to the fact that modifying the ZnO-CuO-SiO_2_ catalyst with catalytically inert WC leads to an increase in the specific surface area of the catalyst (Table 1). In the case of traditional thermal heating, direct dependence of the conversion of ethanol on the catalyst-specific surface area was observed at a low temperature (200 °C), with the maximum difference in the activity being observed. Thus, at this temperature, the conversion of ethanol is 40 and 52.8% for the initial and modified WC-containing samples with a specific surface area of 185 and 245 m^2^/g, respectively. This indicates that the active reaction centers are localized on the surface of the mixed oxide catalyst.

### 2.4. Results of Physicochemical Investigation of the Catalysts

The crystallinity of two samples after catalysis at 200 °C was examined by XRD analysis, and the results are shown in Figure 2. A broad peak between 10 and 50° (2θ region) reveals the presence of amorphous silica together with the peak at 26.45° corresponding to the (101) crystalline phase of silica in two samples after catalysis at 200 °C. The peaks for CuO were found at 36.20 and 61.54°, corresponding to the (111) and (220) cubic modification of copper oxide in these samples. The peak at 35.56° can be ascribed to the (002) reflex from the monoclinic modification of CuO. This assignment is not irrefutable because the peaks at 36.20 and 61.54° can be due to the presence of Cu_2_O as well. The peaks for ZnO were found at 31.69, 34.35, 36.20, 47.49, 56.53 and 67.90°, corresponding to the (100), (002), (101), (102), (110) and (103) hexagonal modification of zinc oxide in these samples. The peaks for WC were found at 31.39, 35.58 and 48.23°, corresponding to the (001), (100) and (101) hexagonal modifications of tungsten carbide in Sample **2**.

The DRIFT spectra measured after CO adsorption on two samples before and after catalysis at 200 °C are shown in Figure 3 and Figure 4. The DRIFT-CO spectrum of the initial three-component CuO-ZnO-SiO_2_ catalyst demonstrates one narrow band at 2196 cm^−1^. This band characterizes both carbonyl on copper cations (Cu^2+^-CO) and on cations of zinc (Zn^2+^-CO) [11,12]. These complexes are unstable, and the band disappears from the spectrum at room temperature in a vacuum. The spectrum of the initial four-component CuO-ZnO-SiO_2_-WC catalyst exhibits two bands. The band at 2203 cm^−1^ with the difference in frequency by 7 cm^−1^ (compared to the band at 2196 cm^−1^) can be ascribed to carbonyls (Cu^2+^-CO) and (Zn^2+^-CO). Alternatively, this band can be attributed to a W^n+^ -CO carbonyl [13]. It is quite difficult to assign the band at 2127 cm^−1^ because it can characterize the carbonyl on copper ions (Cu^+^-CO) inside the SiO_2_ lattice or on the CuO surface [11], as well as oxycarbonyl (Cu^+^(O) -CO) [14]. In addition, a band with such a low frequency of stretching vibrations was found on a Cu/SiO_2_ catalyst modified by sodium [15]. In our case, the role of sodium can be performed by tungsten carbide. Some authors [16] also assigned the band at 2127 cm^−1^ to tungsten carbonyl (W^n+^-CO).

It is seen that there is one band at 2174 cm^−1^ characterizing the stretching vibrations of the C≡O bond in the carbon monoxide molecule adsorbed on copper cations (Cu^+^-CO) in the spectra of both used catalysts.

It should be noted that the intensity of the bands in the DRIFT-CO spectra recorded on the samples after catalysis is an order of magnitude greater than on the original catalysts. This fact can be explained by the fact that during the reaction, copper migrates from the volume to the surface layers. This confirms our assumption that the active reaction centers are localized on the surface of the catalyst samples and explains the difference in the intensities of the DRIFTS-CO bands.

Despite the fact that these catalysts were heated in the presence of H_2_ at 200 °C, the existence of metallic copper on the surface of both used samples was not detected. An explanation of the reason for this can be found below in the TPR-H_2_ section (Section 3.5).

Figure 5 shows the TPR-H_2_ data obtained for the initial samples of CuO-ZnO-SiO_2_ catalysts and the samples after the reaction at 200 °C. For comparison, TPR-H_2_ data for the pure catalyst components, CuO, ZnO and WC, are presented as well.

The uptake of hydrogen on the CuO-ZnO-SiO_2_ catalyst is mainly due to the reduction of the CuO phase. ZnO is not reduced under the experimental conditions, and tungsten carbide actively uptakes hydrogen only after 650 °C. Therefore, all the peaks of hydrogen absorption at 320, 400, 500 and 640 °C can be attributed to the interaction of different copper phases with hydrogen. The total hydrogen uptake (200–800 °C) is 0.811 mmol/g, and the CuO content in the catalyst is 8.13 mmol/g. Thus, the ratio of H_2_/Cu is equal to 0.0997. Obviously, there is no complete reduction of CuO to metallic copper. These data are consistent with the IR and XRD data, which showed the absence of Cu^0^ in the catalysts after the reaction at 200 °C.

For pure CuO, the maximum absorption of H_2_ is observed at 260 °C. For the CuO-ZnO-SiO_2_ catalyst, the first most intense maximum of H_2_ absorption is observed at 320 °C. The interaction of oxides in the prepared catalyst impedes the reduction of Cu^2+^ ions. There is probably an interaction of CuO with ZnO and SiO_2_ during the preparation of the catalyst. An increase in the reduction temperature of the Cu/SiO_2_ catalyst during the introduction of Zn was also observed in the literature [17].

The introduction of tungsten carbide (WC) into the catalyst during the preparation of the catalyst for the possibility of performing efficient microwave heating leads to a significant decrease in the absorption of hydrogen, especially at 320 °C. For the CuO-ZnO-SiO_2_-WC catalyst, the hydrogen absorption in this region decreases by a factor of 24. As a result, the total hydrogen absorption on this catalyst is 0.153 mmol/g. This indicates a strong interaction of copper oxide with tungsten already during the synthesis of the catalyst. Only after the reduction of the CuO-ZnO-SiO_2_-WC catalyst in the TPR runs to 850 °C, the peaks of metallic copper appear in the XRD pattern of this catalyst.

The TPR-H_2_ curves for the catalysts studied after the reaction show almost complete disappearance of the peaks at 320 and 400 °C for both CuO-ZnO-SiO_2_ and CuO-ZnO-SiO_2_-WC catalysts. It is obvious that H_2_ formed during the reaction at 200 °C reduces these components of the catalysts. Because its concentration is higher than in the mixture used in the TPR-H_2_ experiment, and possibly the hydrogen formed in the process of ethanol dehydrogenation has increased activity, compared to molecular H_2_, it can reduce CuO already at a lower temperature (under the experimental conditions, at 200 °C).

## 3. Materials and Methods

### 3.1. Catalyst Preparation and Characterization

Catalysts were prepared by analogy with the preparation of methanol synthesis catalysts by the so-called “wet” method. For this purpose, 200 mL of distilled water was poured into a PAAR-300 metal autoclave, and 5.0 g of ZnO (manufactured by Anton Paar, Germany) was charged under stirring. The aqueous suspension was stirred for 2 h. Then, a portion (5 mL) of Kovelos-20 silica powder was added (SiO_2_ content in water 20 wt %). The suspension was stirred, then ground copper hydroxycarbonate (18.0 g) was added in portions. The resulting mass was heated in an autoclave to a temperature of 90 °C, stirred at this temperature for 2 h and cooled to room temperature. Separation of the precipitate was carried out by filtration. Then, the wet catalyst mass was dried at a temperature of 110 °C for 12 h and calcined with a stepwise temperature rise from room temperature to 450 °C at a rate of 50 °C h^−1^. As a result, a mixed oxide ZnO-CuO-SiO_2_ catalyst was obtained in the form of a highly dispersed powder.

In the preparation of a sample modified with tungsten carbide, a WC nanopowder with a specific surface area of 7 m^2^ g^−1^ and an average particle size of 55 nm obtained from the Institute of Metallurgy (IMET RAS) by the method of plasmochemical synthesis was added to the autoclave during the addition of zinc oxide to the autoclave.

### 3.2. Catalytic Reaction

The dehydrogenation of ethanol was carried out in a flowing quartz reactor (tube with D = 7 mm) at atmospheric pressure and temperatures of 200–400 °C. The reactor was placed in a furnace with electric heating or in a chamber of a household microwave oven “Vigor” with a variable power of 100–1000 W, operating at a frequency of 2.45 GHz. The catalyst charge was 0.5 g (the height of the catalyst bed was 1.2–1.4 cm).

The temperature of the catalyst in the reactor was measured by a thermocouple placed directly in the middle of the catalyst bed and set during operation in the thermal heating mode with a “Thermodat-17” thermoregulator. When operating in the microwave heating mode, the temperature was controlled by the change in the microwave power and by the vertical movement of the reactor with the loaded catalyst in the microwave oven chamber, which has an electromagnetic field gradient along the height.

Heating the catalyst to the reaction temperature was carried out in nitrogen flow supplied from a cylinder through a reducer and a fine adjustment valve. When the desired temperature was reached, the nitrogen flow was turned off, and ethanol was supplied with a syringe dispenser. The mass flow rate of C_2_H_5_OH was 3 h^−1^. Liquid reaction products at the reactor outlet were cooled to −50 °C in a refrigerator and collected in a trap.

Analysis of liquid products was carried out with a 3700 chromatograph manufactured by the Granat NGO (St.-Peterburg, Russia) using a SE-54 capillary column (30 m) and a flame ionization (PID) detector at a temperature of 600 °C (carrier gas was helium).

### 3.3. DRIFTS

DRIFT spectra were collected at room temperature using a NICOLET “Protégé 460” FTIR spectrometer (Madison, WI, USA) with a diffuse-reflectance attachment in the interval 6000–400 cm^−1^ in 4 cm^−1^ increments. Before the spectral measurements, the catalysts loaded into an ampoule with a CaF_2_ window were evacuated to remove physically adsorbed water at a temperature of 350 °C for 2 h (heating rate: 5° min^−1^). Carbon monoxide was used as a probe molecule for the electronic state of elements. Adsorption was carried out at room temperature at an equilibrium pressure of CO 12 Torr.

### 3.4. XRD

Powder X-ray diffraction patterns were recorded with a DRON 3M diffractometer (St.-Peterburg, Russia) using Cu Kα radiation in the Bragg–Brentano reflecting and Debye–Scherrer transmission geometry (λ = 1.54 Å). Diffractograms were collected at room temperature over the range of 10–70° (2θ) with a step of 0.02°, the count time of 0.6 s, and the scan speed of 2° s^−1^. XRD patterns of samples were compared with the JCPDS database of standard compounds (1999—International Centre for Diffraction Data).

### 3.5. TPR-H_2_

Temperature programmed reduction (TPR) measurements were performed in a laboratory constructed flow system built up from the gas purification and supply units, a quartz U-tube reactor, a water vapor trap, and a thermal conductivity detector connected to a data acquisition unit. A water vapor trap was cooled to −100 °C (mixture of ethanol with liquid nitrogen). The detector was calibrated by the reduction of CuO (Aldrich-Chemie GmbH, 99 wt%, Steinheim, Germany) and a mixture of CuO + MgO (to be calibrated at low metal contents) pretreated in Ar flow at 300 °C.

The sample (100–150 μg) was loaded in the reactor, and a thermocouple was mounted inside the reactor close to the sample. In order to minimize the contribution of adsorbed species to the TPR profiles, the sample was pretreated in argon at 300 °C for 1 h prior to the TPR experiment. The TPR run was carried out in a flow of a mixture of 5% of H_2_ in Ar at 30 mL min^−1^ and a heating rate of 10 °C min^−1^ in a temperature range of 20–850 °C. Ar and H_2_/Ar gases with a purity of 99.999% were purified additionally from oxygen using a trap containing a Mn/Al_2_O_3_ catalyst.

## 4. Conclusions

It is shown that the microwave heating of mixed oxide ZnO-CuO-SiO_2_ catalysts modified with additives of tungsten carbide leads to the simultaneous growth of both activity and selectivity compared to traditional thermal heating. It is assumed that with the use of microwave heating, hot zones can appear at the contact points of the tungsten carbide nanoparticles and active centers of the mixed oxide ZnO-CuO-SiO_2_ catalyst, which intensively absorb microwave energy. Such an effect results in a more efficient formation of AA at moderate temperatures. The data from the study of the samples by diffuse-reflectance IR spectroscopy using CO as a probe molecule indicate that the active centers of mixed oxide catalysts are possibly Cu^+^ cations.

An important result with respect to ethanol dehydrogenation technology is that when microwave heating is used for catalysts with additives of conductive carbides, the preheating of raw materials (ethanol) in coils of fire heaters can be excluded, which significantly reduces the energy costs of the process. In addition, during the process, it is possible to produce hydrogen of high purity, which can be used in the development of efficient fuel cells.

## Figures and Tables

**Figure 1 molecules-26-01955-f001:**
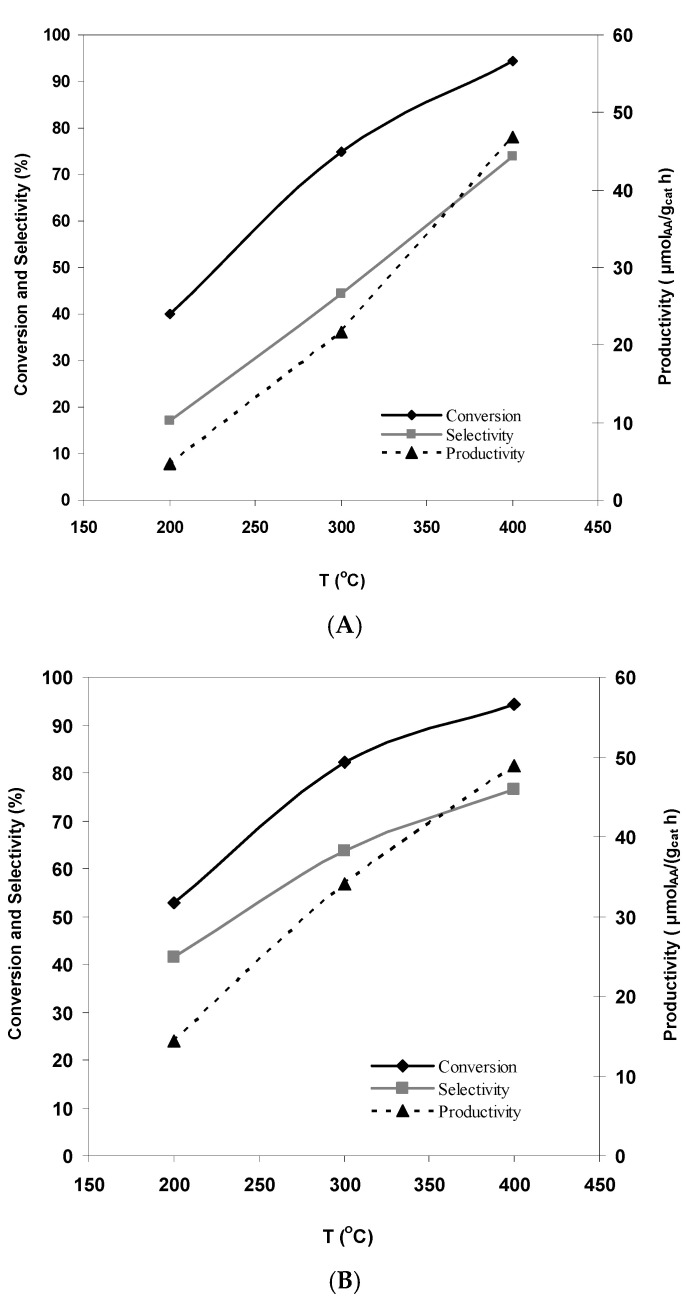
Dependence of the parameters of the process of ethanol conversion on the reaction temperature under thermal (**A**) and microwave (**B**) heating of the catalyst with the composition (wt %) ZnO-22.8; CuO-58.8; SiO_2_-9.2; WC-9.1.

**Figure 2 molecules-26-01955-f002:**
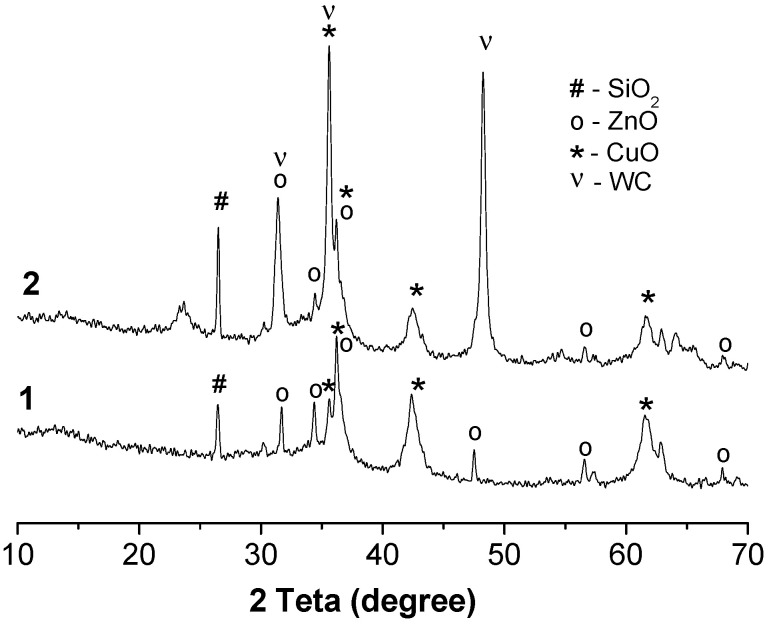
XRD patterns of the CuO-ZnO-SiO_2_ (**1**) and CuO-ZnO-SiO_2_-WC (**2**) catalysts.

**Figure 3 molecules-26-01955-f003:**
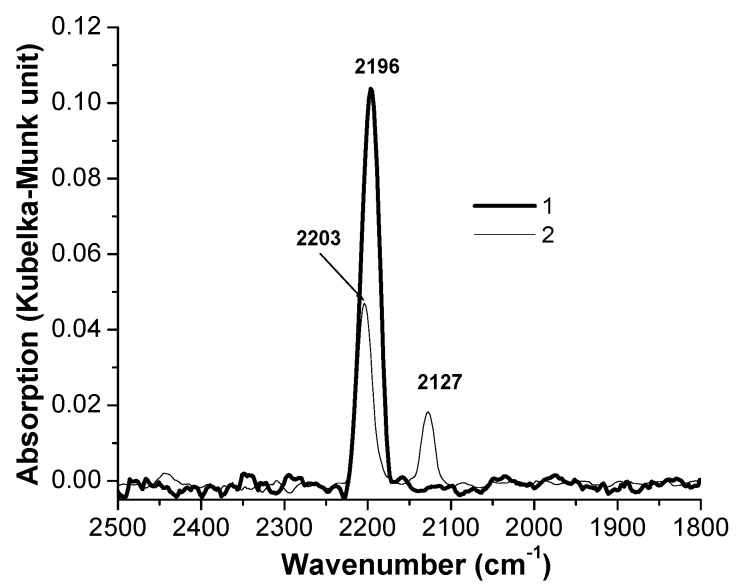
DRIFT-CO spectra measured on the fresh CuO-ZnO-SiO_2_ (**1**) and CuO-ZnO-SiO_2_-WC (**2**) catalysts.

**Figure 4 molecules-26-01955-f004:**
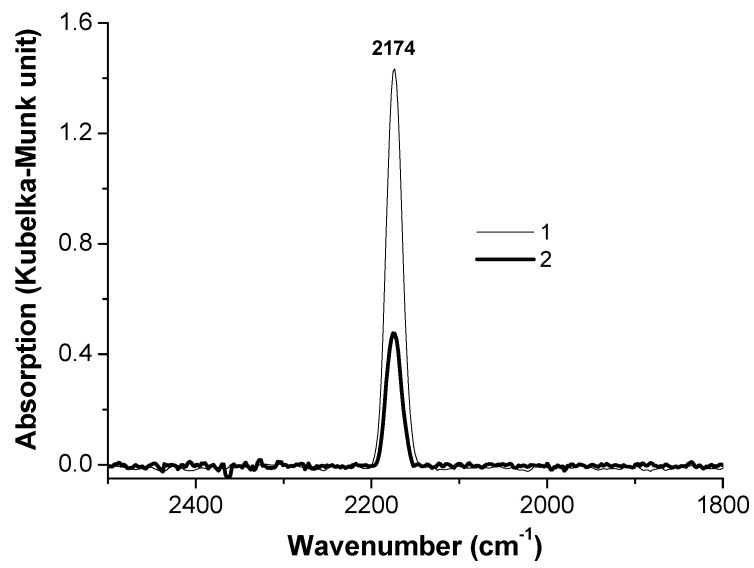
DRIFT-CO spectra of the CuO-ZnO-SiO_2_ (**1**) and CuO-ZnO-SiO_2_-WC (**2**) catalysts after reaction at 200 °C.

**Figure 5 molecules-26-01955-f005:**
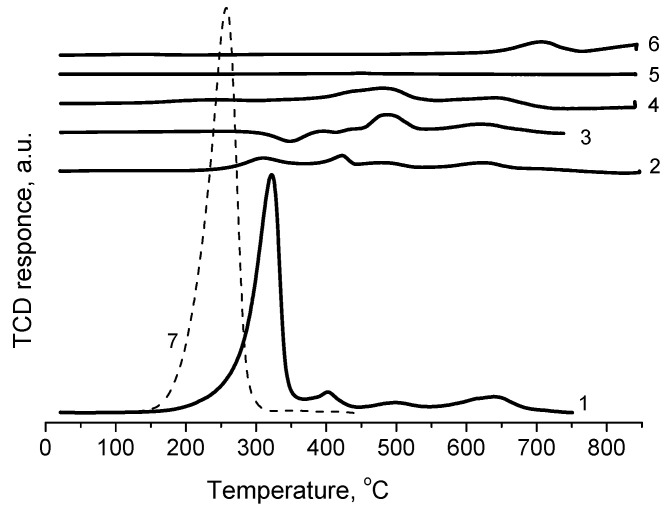
TPR-H_2_ curves for the CuO-ZnO-SiO_2_ and CuO-ZnO-SiO_2_-WC catalysts: **1**—CuO-ZnO-SiO_2_ fresh; **2**—CuO-ZnO-SiO_2_-WC fresh; **3**—CuO-ZnO-SiO_2_ after catalysis; **4**—CuO-ZnO-SiO_2_-WC after catalysis; **5**—ZnO; **6**—WC (scale 1 to 5); **7**—CuO. (All TPR curves are normalized to 1 g; the CuO and tungsten carbide WC curves were normalized to the content of CuO and WC in the catalysts).

**Table 1 molecules-26-01955-t001:** Compositions and specific surface areas of the catalyst samples.

Composition (wt%)	Specific Surface Area (BET), m^2^ g^−1^
ZnO: 25.1; CuO: 64.7; SiO_2_: 10.2	185
ZnO: 22.8; CuO: 58.8; SiO_2_: 9.2; WC: 9.1	245

**Table 2 molecules-26-01955-t002:** Ethanol conversion (C) and selectivity to acetaldehyde (S) over the unmodified ZnO–CuO–SiO_2_ catalyst and the thermodynamic data for different reaction temperatures.

T, °C	Thermal Heating			Thermodynamic Data, Figure 1	
	C, %	S, %	Productivity, μmol_AA_/(g_cat_ h)	C (%)	S (%)
200	34.2	10.8	2.4	73.0	5.2
300	69.1	33.1	14.8	85.0	30.0
400	92.8	72.7	44.0	94.5	77.9

## Data Availability

Not applicable.

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
