# Peer review of "Ethanol to Acetaldehyde Conversion under Thermal and Microwave Heating of ZnO-CuO-SiO_2_ Modified with WC Nanoparticles"

_molecules, 2021, doi:10.3390/molecules26071955_

Round 1
Reviewer 1 Report
This manuscript is a continuation of an article previously published by one of the authors (A. L. Tarasov, “The ethanol to acetaldehyde conversion activated by microwave radiation”, Russian Chemical Bulletin, International Edition, Vol. 67, No. 8, pp. 1390—1393, August 2018). The present manuscript deals with the study of the catalytic system, ZnO-CuO-SiO2 modified with WC, and the Tarasov's article studies the ZnO—CuO—SiO2 oxide catalysts modified with additives of Nb and Ta carbides. Both catalytic systems are devoted to the ethanol to acetaldehyde conversion activated by microwave radiation. This manuscript includes characterization results like TPR-H2, XRD, DRIFTS of adsorbed CO), and Tarasov's article only presents catalytic activity results. So, I recommend this article to make a major revision.
- Authors are suggested to include a paragraph in the introduction highlighting the differences of the present manuscript concerning Tarasov's.
- If carbides are inert for this reaction, why study the effect of the type of carbide (i.e., W, Nb, and Ta carbides)?
- Please, remove Fig.1 and refer to the thermodynamic study reported by Tarasov.
- Authors claim that "studies of copper-containing catalysts indicate that copper (Cu°) in the metallic state formed during the reduction of CuO is an active catalyst" (lines 37-39). Why did the authors not carry out a prior reduction of their catalysts?
- Authors claim in their conclusions that “data from the study of the samples by physicochemical methods indicate that the active centers of mixed oxide catalysts are presumably Cu+ cations.” (lines 358-360). However, they did not detect the presence of Cu2O by their characterization results; therefore, there is no support for the latter conclusion.
Author Response
Reviewer 1 comments
First, we would like to thank the reviewer for the careful analysis of our manuscript and valuable comments that helped us to improve the paper. Below we provide responses to the reviewer comments.
Comment 1
This manuscript is a continuation of an article previously published by one of the authors (A. L. Tarasov, “The ethanol to acetaldehyde conversion activated by microwave radiation”, Russian Chemical Bulletin, International Edition, Vol. 67, No. 8, pp. 1390—1393, August 2018). The present manuscript deals with the study of the catalytic system, ZnO-CuO-SiO2 modified with WC, and the Tarasov's article studies the ZnO—CuO—SiO2 oxide catalysts modified with additives of Nb and Ta carbides. Both catalytic systems are devoted to the ethanol to acetaldehyde conversion activated by microwave radiation. This manuscript includes characterization results like TPR-H2, XRD, DRIFTS of adsorbed CO), and Tarasov's article only presents catalytic activity results. So, I recommend this article to make a major revision.
- Authors are suggested to include a paragraph in the introduction highlighting the differences of the present manuscript concerning Tarasov's.
Response:
We added the following paragraph in the Introduction: “This manuscript is a continuation of an article previously published by one of the authors (A. L. Tarasov) [10]. The present manuscript deals with the study of the catalytic system, ZnO-CuO-SiO2 modified with WC, and the previous article has been focused on the study of the ZnO—CuO—SiO2 oxide catalysts modified with additives of Nb and Ta carbides. Both catalytic systems are used in the ethanol to acetaldehyde conversion activated by microwave radiation. This manuscript additionally includes extensive characterization results, including TPR-H2, XRD, DRIFTS of adsorbed CO, whereas the previous article only presented catalytic activity data.”
Comment 2
- If carbides are inert for this reaction, why study the effect of the type of carbide (i.e., W, Nb, and Ta carbides)?
Response: Actually, we studied other additives except for carbides of W, Nb, Ta, such as carbon materials, V-Mo oxide systems, SiC, etc. with the goal to identify the material that (1) absorb very well microwave radiation, (2) does not provide any catalytic activity in the chosen reaction and (3) is quite stable under the experimental conditions of the catalytic reaction. WC was found to fit all the three criteria, moreover, it was a better microwave absorber than TaC and NbC, therefore, it was chosen for the detailed study.
Comment 3
- Please, remove Fig.1 and refer to the thermodynamic study reported by Tarasov.
Response: We removed Fig. 1 and gave a reference to the previous work.
Comment 4
- Authors claim that "studies of copper-containing catalysts indicate that copper (Cu°) in the metallic state formed during the reduction of CuO is an active catalyst" (lines 37-39). Why did the authors not carry out a prior reduction of their catalysts?
Response: The reaction mixture used in the studies is a rather strong reducing agent, so no preliminary reduction with hydrogen was required. Also, the state of copper was anyway controlled by IR spectroscopy of CO.
Comment 5
- Authors claim in their conclusions that “data from the study of the samples by physicochemical methods indicate that the active centers of mixed oxide catalysts are presumably Cu+ cations.” (lines 358-360). However, they did not detect the presence of Cu2O by their characterization results; therefore, there is no support for the latter conclusion.
Response: It is seen from Fig. 4 that there is one band at 2174 cm-1 characterizing the stretching vibrations of the C≡O bond in the carbon monoxide molecule adsorbed on copper cations (Cu+-CO) in the spectra of both used catalysts. Therefore we made the conclusion about the participation of Cu+ species in the reaction. However, we revised the sentence in Conclusions to make it not so straight-forward: “The data from the study of the samples by diffuse-reflectance IR spectroscopy using CO as a probe molecule indicate that the active centers of mixed oxide catalysts are possibly Cu+ cations.”
Reviewer 2 Report
The authors developed an exhaustive work that is well presented. In my opinion, the manuscript is appropriate, and deserves to be published in Molecules. I just would like to propose i) a deeper explanation of the role of WC (considering its thermal conductivity) and why the authors have chosen it in comparison with alternative materials and their effects; ii) the addition of a kinetic profile (vs. time); and some comments regarding the reuse of the catalyst. The English style could also be revised.
Author Response
Reviewer 2 comments
First, we would like to thank the reviewer for the careful analysis of our manuscript and valuable comments that helped us to improve the paper. Below we provide responses to the reviewer comments.
Comment 1
The authors developed an exhaustive work that is well presented. In my opinion, the manuscript is appropriate, and deserves to be published in Molecules. I just would like to propose i) a deeper explanation of the role of WC (considering its thermal conductivity) and why the authors have chosen it in comparison with alternative materials and their effects; ii) the addition of a kinetic profile (vs. time); and some comments regarding the reuse of the catalyst. The English style could also be revised.
Response: Actually, we studied other additives except for carbides of W, Nb, Ta, such as carbon materials, V-Mo oxide systems, SiC, etc. with the goal to identify the material that (1) absorb very well microwave radiation, (2) does not provide any catalytic activity in the chosen reaction and (3) is quite stable under the experimental conditions of the catalytic reaction. WC was found to fit all the three criteria, moreover, it was a better microwave absorber than TaC and NbC, therefore, it was chosen for the detailed study. Therefore, we added the following paragraph: “We studied various additives except for carbides of W, Nb, Ta, such as carbon materials, V-Mo oxide systems, SiC, etc. with the goal to identify the material that (1) absorb very well microwave radiation, (2) does not provide any catalytic activity in the chosen reaction and (3) is quite stable under the experimental conditions of the catalytic reaction. WC was found to fit all the three criteria, moreover, it was a better microwave absorber than TaC and NbC, therefore, it was chosen for the detailed study.”
Concerning the kinetic dependences, we do hope that the data on the conversion, selectivity and space-time yield (productivity expressed in micromoles of the product per gram of catalyst per hour are sufficient to demonstrate the effect of the WC additive and the microwave heating in comparison with the thermal process.
The English language was edited.
Round 2
Reviewer 1 Report
The changes made by the authors have substantially improved the manuscript, therefore, it should be accepted for publication.